# Exploratory and Descriptive Comparison Using the WAIS-IV and LSC-SUA of the Cognitive Profile of Italian University Students with Reading Comprehension Disorder (RCD) vs. Decoding Difficulties—Dyslexia

Daniela D'Elia, Luna Carpinelli  and Giulia Savarese *

Department of Medicine, Surgery and Dentistry, University of Salerno, 84081 Baronissi, Italy
* Correspondence: gsavarese@unisa.it; Tel.: +39-089965079

**Abstract: Background:** The "Guidelines for the Management of Specific Learning Disorders" provide clear diagnostic and evaluative guidance on Reading Comprehension Disorder (RCD), as suggested by the DSM-5. The present study investigated the relationship between cognitive abilities in university students with RCD compared to students with Decoding Difficulties—i.e., dyslexia (DD)—and examined possible diagnostic procedures for RCD in young adults. **Methods:** Twenty university students from the University of Salerno "Disabilities/SLD Help Desk" participated. The sample was divided into the RCD Group (10) and DD Group (10). They were administered (a) the Wechsler Adult Intelligence Scale—Fourth Edition (WAIS-IV) to assess their cognitive profile; and (b) the LSC-SUA-Reading tests for the assessment of text comprehension, writing, and calculation skills. **Results:** With regard to their cognitive profile, the DD Group had a lower mean of FSIQ (91.90 ± 5.82 vs. 92.50 ± 2.41). The RCD Group showed higher means in the subscales of CPI (94.80 ± 5.77), WMI (91.80 ± 78.80), and PSI (101.20 ± 6.20). Text comprehension assessment showed high averages in the DD Group (Track A = 11.50 ± 1.08; Track B = 11.40 ± 0.95). **Conclusions:** A valid psychodiagnostic model to examine reading comprehension skills for adults must assess the interaction between higher-level linguistic and cognitive processes in order to best define the pathways of skill enhancement.

**Keywords:** specific learning disorders; reading comprehension disorder; decoding difficulties; cognitive profile



## 1. Introduction

In Italy, the number of university students with dyslexia and other specific learning disabilities (SLD) is growing; in 2020–2021, there were 19,616 students with SLD, an increase of 22% compared to the previous academic year. According to research carried out by the national agency for the evaluation of the university and research system in Italy [1], 54% of university students with SLD chose sociohumanistic disciplines, 28.7% chose scientific areas, and 9.4% chose health. Approximately 13,127 students with SLD were enrolled in three-year degree courses, 1507 in Master's degree courses, and 1416 in single-cycle courses. A total of 90 out of 98 universities participated in the survey: 76.7% offered specific guidance services (before, during, and after the study course), and 68.7% provided support for teaching, including specialized or peer tutoring and the provision of books in accessible formats. Clearly, understanding and recognizing SLD at university is essential to enable students to access tutoring services and to be able to complete their studies to the best of their ability [1].

In February 2022, the Italian National Institute of Health published the document "Guidelines on the Management of Specific Learning Disorders. Update and Additions" [2], in which the criteria for evaluating SLDs were defined, recommendations for diagnosis were given, and indications for the most effective treatments were discussed. There was

a focus on Reading Comprehension Disorder (RCD), whose diagnostic and assessment indications had remained unresolved in the Consensus Conference [3], but the need to provide precise indications had emerged with the publication of the DSM-5 [4] and with the redefinition of RCD.

In the DSM-5, dyslexia is coded as a "Specific Learning Disorder with reading impairment" (code 315.00) [4] and includes, in addition to difficulties in reading accuracy, difficulties in reading speed and text comprehension. Thus, the DSM-5 implicitly takes a stand in relation to the ambiguous differentiation between problems involving reading, such as decoding and reading comprehension, present in the previous version of the DSM and ICD-10 [5].

The DSM-5 has introduced a uniform diagnostic label for SLDs but requires that the area that is deficient be specified by means of descriptors. With regard to RCD, it is stated that it is necessary to specify whether this concerns decoding or text comprehension. If the learning disorder concerns reading comprehension, the DSM-5 describes it as a difficulty in understanding the meaning of what is read in the presence of a good decoding ability, which implies a difficulty in understanding the sequence, the relationship between the information contained in the text, and what is not explicitly said (inferences); in other words, the deeper meaning of what is read.

In line with the DSM-5, the new SLD guidelines make it explicit that text comprehension difficulties must not be a consequence of a decoding disorder and require the use of standardized and independent tests. To this end, among the questions in the new document is a special emphasis on what criteria and diagnostic procedures should be used to ascertain RCD in young adults.

SLDs have a character of persistence due to their neurobiological basis [6–8]; therefore, it is to be expected that, even in adulthood, there will be a large population of subjects with difficulties in functioning in certain tasks and activities of daily life and in the school/university or professional sphere who were not diagnosed in childhood [9]. In contrast to the multiple and comparable instruments available for the developmental stage, tests for adults are scarce and the literature is limited.

The new guidelines, with reference to which instruments to use for the diagnosis of SLD in adults, specify how it is essential to consider how text comprehension disorder can be studied and thus assessed as a disorder with characteristics quite distinct from decoding disorder. In particular, the research suggests that among the distinctive differences are those attributable to underlying cognitive abilities and processes [9].

In line with the recommendations of the new Consensus Conference [2], the present study had the following objectives: (a) describe the cognitive profile assessed with the Wechsler Adult Intelligence Scale—Fourth Edition (WAIS-IV) [10] in adult students with RCD in comparison with students with Decoding Disorder (DD); and (b) examine, also in the Italian context, which diagnostic procedures can validly contribute to ascertaining RCD in young adults.

## 2. Materials and Methods

### 2.1. Procedure

The assessment was carried out at the "Disabilities/SLD Help Desk" of the University of Salerno (Italy). This service is aimed at any student who:

- Is experiencing difficulties in studying and/or may have an SLD that has not yet been recognized;
- Is already certified as having an SLD and who requires support to update their diagnosis/certification or specific guidance on study methods.

The service has established a protocol with the Local Health Authority (LHA) which provides for a psychologist expert in learning psychopathology of the University service to administer the learning assessment battery to the students, defining their profiles, and then for the LHA multidisciplinary team to complete the clinical assessment (also considering a possible differential diagnosis).

The assessment process consists of 3–4 meetings. During the first meeting, an anamnesis and analysis of the needs are carried out: the reason for the student's request for counselling (first assessment or clinical update) is defined and information about the student's developmental and functional history is collected. In addition, the Vinegrad Plus questionnaire [11], consisting of 26 questions with dichotomous answers (yes/no), is administered, which allows for the investigation of current conditions and abilities in everyday tasks that engage read-to-write and abilities related to automatable aspects of social skills and language components [12].

In the second and third meeting, the Reading, Writing, and Calculation tests of the LSC-SUA [12] and the WAIS-IV [10] are administered, which, depending on the student's difficulties and timeframe, may require another meeting.

The last meeting is dedicated to the "restitution" phase, i.e., sharing the results and indications of a pathway that may involve either sending the student to the LHS to complete the diagnosis certification or access to the "Disabilities/SLD Help Desk".

### 2.2. Materials

1.  Wechsler Adult Intelligence Scale—Fourth Edition (WAIS-IV) [10] is employ for assessing the cognitive profile and providing a measure of general intellectual functioning (FSIQ) score representative of general intellectual ability. The scale is structured based on four composite scores which measure specific cognitive domains: Verbal Comprehension Index (VCI); Perceptual Reasoning Index (PRI); Working Memory Index (WMI); and Processing Speed Index (PSI). In addition, in accordance with the study by Cornoldi et al. [13], we used two additional composite indices: General Ability Index (GAI, consisting of the sub-tests VCI and PRI) and Cognitive Proficiency Index (CPI, consisting of WMI and PSI);
2.  LSC-SUA [12] is a battery of tests to evaluate the fundamental aspects of reading, writing, text comprehension, and calculation in university students and adults. The battery analyzes barriers and facilitators that can hinder/foster student educational success with SLD. The standardization of the battery was carried out in collaboration with several Italian universities, including the University of Salerno, in order to have a sample as representative as possible of the different cultural realities. The tests are divided into 4 areas: (1) Reading: reading of a passage, reading of words and not words, lexical decision in articulatory suppression; (2) Comprehension of the text: Track A and Track B; (3) Writing: dictation of words in normal conditions and in articulatory suppression conditions; dictation of passage; writing of numbers in letters in normal conditions and in articulatory suppression conditions; (4) Calculation: dictation of numbers, reading of numbers, calculation in mind, arithmetic facts, approximate calculation, and transcription of digits. To complete the battery, a self-awareness questionnaire (Vinegrad Plus) concerning their difficulties is to be administered to students [11].

In order to achieve the objectives of this study, we analyze the results of the Comprehension of the text. This assessment includes two tracks in the Italian language: Track A, "L'Atlantide dell'Adriatico", adapted from "Inseguendo l'Isola Valbruna l'Atlantide dell'Adriatico" by Alessandra Leardini (La Repubblica, 19 August 2009), integrated with documents present on the web; and Track B, an additional test, "Cronaca di una tragedia annunciata", adapted from an in-depth analysis of "Focus" (http://dentroilvajont.focus.it/il-dopo-vajont.html accessed on 31 December 2022) and from documents present on the web.

### 2.3. Participants

Twenty students aged between 20 and 25 were selected (12 males; $M = 21.5$; $SD = 1.57$) who have had access to the service of the "Disabilities/SLD Help Desk" of the University of Salerno (Italy) during the academic years 2020–2021. In accordance with the DSM-5 and ICD-10 diagnostic criteria) for SLD, students with the following were excluded from the

reference sample: Intellectual disability (WAIS-IV scale scores within the normal range); neurological disorder, trauma or illness; sensory disorder; psychosocial disadvantage; deprivation due to inadequate educational environment.

The sample was divided into two groups: n°10 students with a diagnosis of RCD (RCD Group) and n°10 with Decoding Difficulties—i.e., dyslexia (DD Group).

Inclusion criteria for DD Group:

In the case of the DD Group, the recommendations for the assessment of reading decoding ability as indicated in the Italian National Institute of Health Guidelines [2] were followed:

- Standardized tests were administered for adults reading aloud passages, words, and non-words (LSC-SUA). Both speed and accuracy were measured (Recommendation 8.1);
- Tests were also administered to assess the ability to comprehend written text (LSC-SUA) considering accuracy, with tests appropriate to age and schooling and with demonstrated clinical validity (Recommendation 8.2);
- Tests were administered to qualify the diagnosis and profile, using subtests of the WAIS-IV, which also assessed verbal working memory and speed of information processing (Recommendation 8.3).

Inclusion criteria for RDC Group:

In order to identify students with RCD, in line with the most recent indications of the Italian National Institute of Health Guidelines [2] on SLD, the following recommendations were used:

- Presence of a cognitive profile in the normal range (FSIQ or GAI $\geq$ 85) and good decoding ability;
- A cut-off equal to the 10th percentile was used as a psychometric indicator in the interpretation of comprehension test results (Recommendation 2.1);
- Two tests were used to assess comprehension of the text (LSC-SUA Tracks A and B);
- Subjects whose performances were clinically significant in both tests were considered part of the RCD sample (Recommendation 2.2);
- Independent tests of decoding and text comprehension were used (Recommendation 2.3);
- Word list and non-word list reading tests were used in order to exclude the presence of a problem in reading as decoding as well (Recommendation 2.4);
- The profile of the disorder was deepened by analyzing performances in tests assessing vocabulary (receptive and/or expressive), syntactic and grammatical comprehension, and basic processes such as verbal working memory, with subtests of the WAIS-IV (Recommendation 2.5);
- A measure of non-verbal reasoning (some subtests of the WAIS-IV) was used in order to ensure that low competence in text comprehension is not dependent on more general difficulties (Recommendation 2.6).

For both subgroups, students were selected on the basis of their score on the Vinegrad Plus (cut-off = 11) [12].

### 2.4. Data Analyses

The Statistical Package for the Social Sciences, version 22 (SPSS Inc., Chicago, IL, USA) was used on all data analyses. The t-test was used to investigate differences between independent samples. In addition, a descriptive comparison between the mean (M) and standard deviation (SD) was made to compare the differences between the profiles of the two groups. Pearson's correlation analysis was performed to investigate the relationship between subscales of WAIS IV and LSC-SUA tests and different diagnosis.

## 3. Results

### 3.1. Cognitive Profile

Table 1 shows the differences in the mean scores obtained on the WAIS IV scales by the two reference groups. As can be verified, the DD Group had a lower mean FSIQ than the RCD Group (M = 91.90, SD = 5.82 vs. M = 92.50, SD = 2.41; *p* = 0.767). The DD Group had higher means on the WAIS-IV subscales regarding GAI (M = 106.30, SD = 7.05; *p* = 0.000), VCI (M = 104, SD = 8.64; *p* = 0.000) and PRI (M = 106, SD = 6.18; *p* = 0.443) than the RCD Group. The RCD Group showed higher means in CPI (M = 94.80, SD = 5.77; *p* = 0.002), WMI (M = 91.80, SD = 78.80; *p* = 0.013) and PSI (M = 101.20, SD = 6.20; *p* = 0.000) than the DD Group.

**Table 1.** Mean differences in WAIS-IV scale scores: comparison between groups.

|  | Group | M | SD | *t* | df | *p* |
|---|---|---|---|---|---|---|
| **FSIQ** | DD | 91.90 | 5.82 | −0.301 | 18 | 0.767 |
|  | RCD | 92.50 | 2.41 |  |  |  |
| **GAI** | DD | 106.30 | 7.05 | 5.075 | 18 | 0.000 |
|  | RCD | 93.00 | 4.34 |  |  |  |
| **CPI** | DD | 66.70 | 23.23 | −3.711 | 18 | 0.002 |
|  | RCD | 94.80 | 5.77 |  |  |  |
| **VCI** | DD | 104.00 | 8.64 | 7.744 | 18 | 0.000 |
|  | RCD | 81.80 | 2.74 |  |  |  |
| **PRI** | DD | 106.00 | 6.18 | 0.785 | 18 | 0.443 |
|  | RCD | 104.00 | 5.16 |  |  |  |
| **WMI** | DD | 78.80 | 12.90 | −2.758 | 18 | 0.013 |
|  | RCD | 91.80 | 7.45 |  |  |  |
| **PSI** | DD | 76.80 | 7.45 | −8.029 | 18 | 0.000 |
|  | RCD | 101.20 | 6.07 |  |  |  |

WAIS-IV Index: Total Intelligence Quotient (TIQ); Verbal Comprehension Index (VCI); Perceptual Reasoning Index (PRI); Working Memory Index (WMI); Processing Speed Index (PSI); General Ability Index (GAI); Cognitive Proficiency Index (CPI); t-test value (*t*); degree of freedom (df); level of significance (*p*).

### 3.2. Reading Comprehension

From the evaluation of the LSC-SUA tests, the scores obtained on Tracks A and B were recorded. Three scores were calculated for each track: a total score (TOT) of text comprehension; a score of the inferential/general questions (IG); and a score of the specific questions (S).

The analysis of the mean total scores obtained in Tracks A and B (see Table 2) showed high averages (*p* = 0.000) in the DD Group (Track A: M = 11.40, SD = 0.96; Track B: M = 11.50, SD = 1.08) compared to the RDC Group (Track A: M = 5.80, SD = 0.63; Track B: M = 5.30, SD = 0.67). The IG score shows higher averages (*p* = 0.000) in the DD Group than in the RCD Group on both Track A (M = 5.20, SD = 1.31 vs. M = 2.70, SD = 0.48) and Track B (M = 5.90, SD = 0.87 vs. M = 2.50, SD = 0.07). The score of the specific questions (S) also shows this discrepancy by presenting different performances of the two groups with higher averages (*p* = 0.000) in the DD Group (Track A: M = 5.70, SD = 0.87; Track B: M = 6.20, SD = 0.63).

In addition, the scores of the two study groups were compared with the synthetic normative data for the standardisation of the LSC-SUA [12]. The normative data for Track A were collected from a total sample of 562 students, 344 (61%) females and 218 (39%) males, 273 (49%) attending a degree course in the humanistic-social area, and 289 (51%) in the technical-scientific area. The normative data of the second comprehension test (Track B) were collected from a sample of 451 students, 264 (58%) females and (42%) 187 males, 56% attending a degree course in the technical-scientific area, and 44% in the humanistic–social area [12].

**Table 2.** Mean differences in LSC-SUA: comparison between groups in Tracks A and B.

|  | Group | M | SD | *t* | df | *p* |
|---|---|---|---|---|---|---|
| **Track A (TOT)** | **DD** | 11.40 | 0.96 | 15.393 | 18 | 0.000 |
|  | **RCD** | 5.80 | 0.63 |  |  |  |
| **Track A (IG)** | **DD** | 5.20 | 1.31 | 9.553 | 18 | 0.000 |
|  | **RCD** | 2.70 | 0.48 |  |  |  |
| **Track A (S)** | **DD** | 5.70 | 0.87 | 6.539 | 18 | 0.000 |
|  | **RCD** | 2.80 | 0.78 |  |  |  |
| **Track B (TOT)** | **DD** | 11.50 | 1.08 | 15.336 | 18 | 0.000 |
|  | **RCD** | 5.30 | 0.67 |  |  |  |
| **Track B (IG)** | **DD** | 5.90 | 0.87 | 5.637 | 18 | 0.000 |
|  | **RCD** | 2.50 | 0.07 |  |  |  |
| **Track A (S)** | **DD** | 6.20 | 0.63 | 11.535 | 18 | 0.000 |
|  | **RCD** | 3.10 | 0.56 |  |  |  |

LSC-SUA: Total score of text comprehension (TOT); score of the inferential/general questions (IG); score of the specific questions (S); *t*-test value (*t*); degree of freedom (df); level of significance (*p*).

Table 3 shows that both the DD Group and the RCD Group have different performances compared to the normative group and especially the RCD Group shows a clear lowering of TOT, IG, and S score averages.

**Table 3.** Comparison between DD Group, RCD Group, and Normative Group in Tracks A and B (LSC-SUA).

|  |  | Track A (TOT) | Track B (TOT) | Track A (IG) | Track B (IG) | Track A (S) | Track B (S) |
|---|---|---|---|---|---|---|---|
| **DD Group** | **Mean** | 11.40 | 11.50 | 5.20 | 5.90 | 5.70 | 6.20 |
|  | **SD** | 0.96 | 1.08 | 1.31 | 0.87 | 0.87 | 0.63 |
| **RCD Group** | **Mean** | 5.80 | 5.30 | 2.70 | 2.50 | 2.80 | 3.10 |
|  | **SD** | 0.63 | 0.67 | 0.48 | 0.07 | 0.78 | 0.56 |
| **Normative Group** | **Mean** | 10.19 | 10.97 | 5.54 | 4.72 | 4.53 | 6.25 |
|  | **SD** | 2.53 | 2.17 | 1.49 | 1.19 | 1.62 | 1.40 |

*3.3. Correlation*

The correlation analysis carried out between the diagnosis variable and the scales of the WAIS IV showed a negative correlation ($p < 0.01$) with the GAI ($r = -0.767$), VCI ($r = -0.877$) and positive correlation ($p < 0.01$) with CPI ($r = 0.658$) and PSI ($r = 0.884$) WMI ($p < 0.05$; $r = 0.545$).

Significant correlations ($p < 0.01$) between the diagnosis variable and the tests of the LSC-SUA emerged between the total scores of tests A ($r = -0.964$) and B ($r = -0.964$), with IG scores for Track A ($r = -0.914$) and B ($r = -0.799$) and S scores for Track A ($r = -0.839$) and B ($r = -0.939$).

**4. Discussion**

On the basis of the results obtained from our study, we highlight the following in reference to the research objectives.

*(a) Describe the cognitive profile assessed with the Wechsler Adult Intelligence Scale—Fourth Edition (WAIS-IV) in adult students with RCD in comparison with students with Decoding Disorder (DD).*

Specific reading comprehension disorder refers to reading difficulty that does not concern the ability to decode text (reading remains adequately fluent and correct), but the ability to grasp meaning effectively. In the DSM-5 [4], both reading disorders (decoding vs. comprehension) fall under a single diagnostic category (code 315), so the first focus of this

study was to analyze both the cognitive and performance characteristics of the two clinical sub-profiles of dyslexia.

The ability to understand a written text is a multi-componential process that involves different linguistic and cognitive processes that operate on the text and interact not only with each other, but also with basic knowledge and metacognitive knowledge.

A fundamental role is played by working memory and prior knowledge relating both to the language and the subject matter and the processes of making connections between the different parts of the passage in order to have a coherent representation of the text. Therefore, a good psychodiagnostic model to examine reading comprehension skills for adults must assess the interaction between higher level linguistic and cognitive processes [12].

The structure of the WAIS-IV lends itself well to this evaluation, favoring the possibility of capturing, within an SLD with impaired reading, specificities between subjects with Decoding Disorder and subjects with Reading Comprehension Disorder. In our study, we have seen how, from the comparison of the average scores recorded on the WAIS-IV scales, the DD Group achieved higher mean than the group with RCD in general abilities (GAI), in verbal comprehension (VCI), and processing speed (PRI) scales.

In contrast, the RCD Group showed higher averages in the score of the general intellectual functioning (FSIQ), in cognitive competence (CPI), in working memory (WMI), in processing speed (PSI), and in verbal comprehension (VCI).

Many researchers have already shown that SLDs are associated with poor performance in working memory and processing speed [14,15]; however, most of these studies referred to dyslexia only in terms of decoding deficits, suggesting the use of the general abilities (GAI) and not the general intellectual functioning (FSIQ) as an intellectual ability index may be preferable in order to correctly assess the presence of an SLD [16].

The peculiarities and differences in the cognitive profile of the two subtypes of reading disorders make it necessary to examine the various domains as a whole. The WAIS-IV, which is characterized by a greater attention to different indices than previous versions, is even more useful than previously for the diagnosis and study of SLDs [17].

SLDs, therefore, involve a specific domain of ability, leaving general intellectual functioning intact [2]. In fact, the disorder affects a specific skill (reading, writing, calculation) while preserving cognitive functioning, albeit with peculiarities. According to various studies, from the analysis of cognitive functioning of subjects with SLD, difficulties are often observed in processing speed and/or working memory, whereas visual–perceptual reasoning often appears to be an area of strength. The discrepancy between these areas often means that the FSIQ (WAIS-IV) cannot be interpreted, precisely because it becomes an average between numbers that are too different from each other and not indicative of homogeneous functioning. Therefore, when interpreting the test, it is also important to calculate the General Ability Index (GAI), which allows the assessment of general intelligence, excluding the auditory short-term memory and working memory tests and the processing speed tests.

Despite having an excellent—often above average—score of intelligence in some areas, in the protocols of the tests conducted on young adults with SLD we will hardly find a medium high FSIQ, as the objective parameters with which it is calculated are limiting and penalizing for those with SLD, who have specific skills and their performance in testing is very often uneven.

In our study, the DD Group, has a better performance than the RCD Group in the subtest that make up the Verbal Comprehension Index (VCI), including "Vocabulary". In a study of Cavalli et al. [18] it was found that dyslexics do not differ from controls in the Comprehension test, while they have a better performance in the Vocabulary task.

This study showed that college students with evolutionary dyslexia possess good vocabulary skills. Therefore, subjects with dyslexia could use this ability as a strategy to compensate for reading difficulties and succeed in understanding the text.

In any case, data from adult literature are generally scarce and relate almost exclusively to the English language context, so the generalization to the Italian reality should be carried out with caution, taking into account the different spelling characteristics of the two languages [19].

*(b) Examine, also in the Italian context, which diagnostic materials can help to ascertain RCD in young adults.*

The tracks of the LSC-SUA have been identified as a valuable tool for diagnostic evaluation, including discriminating the performance of students with RCD and DD. In fact, from the analysis of the average scores obtained on the comprehension of the LSC-SUA tracks, the DD Group had higher average scores than the RCD Group on both Track A and B in all the characterizing tests. In addition, the average scores to inferential/general questions (IG) in the RCD Group were lower, in line with several studies that have shown that significant differences can be found when the task requires inferential processes [20–23].

This seems to be particularly interesting considering the two parts of the LSC-SUA are considered easily readable by students with a secondary school degree [24], while academic texts are generally longer, more complex, and characterized by abstract and technical language in new fields. However, the fact that such passages also contain the distinction between inferential and specific questions seems an important added value. For the text to be comprehensible, a reader must make inferences in the text, linking the elements of the text with the intention of building a mental logic of it.

Inferences are usually triggered by implicit or even absent elements in the text and therefore must be mentally generated by the reader to understand it. The ability to make textual inferences, therefore, is also an excellent indicator of an experienced reader, because it is necessary to make inferences for a full understanding of the text [12].

This highlights what is already present in the literature: namely, that students with RCD have greater difficulty processing texts, further highlighting the importance of specific evaluation of the higher processes involved.

An analysis of the literature of studies that have dealt with SLD in adulthood, also using the WAIS (version R, III or IV), with regard to the differences in cognitive profile between the various subtypes of SLD reveals interesting results also in comparison with the present study. Specifically, the study by Callens et al. [25], with reference to the global cognitive profile analysis of adult subjects with dyslexia, found the presence of deficient performances in numerous tasks administered, particularly in the WAIS tests, which involve the speed of information processing and the retrieval of verbal information from long-term memory. Moreover, in Ransby's study [26], subjects with developmental dyslexia underperformed the chronological peer group in phonological processing, processing speed, working memory, and vocabulary. In the study of Godoy de Oliveira et al., [27] in which a group of dyslexics was compared to a control group, it was found that dyslexics perform below 2 standard deviations (<2 SD) in visual and auditory lexical decision making and phonological awareness tests, while total intelligence is greater than 80 (FSQI > 80).

The study of Wiseheart et al. [28] on a group of adults with dyslexia (vs. control group) demonstrated that syntactic deficits, assessed with sentence comprehension of various types (active/passive; relative etc.) in dyslexic adults are conditioned by working memory and word reading ability. Difficulties with sentence comprehension in dyslexia do not reflect an underlying deficit in syntactic processing per se; rather, they reflect yet another manifestation of difficulty in retaining and verbally manipulating information encoded in working memory.

It is certainly noticeable that the above-mentioned studies mainly a generic reference to "Reading Disorder" without specifying whether it is in decoding (which in any case is more studied) or in comprehension. Nevertheless it seems useful to present these analyses because, on the whole, they seem to highlight the importance of administering a broad battery of neuropsychological tests in order to make an accurate differential diagnosis and design interventions in adult subjects.

With regard to the differences in the cognitive profile between the various subtypes of SLD, it is interesting to note what emerged from a multicentre study [29] that progressively led to the establishment of a database relating to almost 2000 cases diagnosed with SLD assessed with the WISC-IV battery (a version of the WAIS for subjects aged between 6 and 16.11 years). Among the main results of this survey, it is considered useful for the purposes of our study to highlight that (a) the factorial structure of intelligence assessed by means of the WISC-IV can be considered, in the first instance, as overlapping with that of subjects with typical development, but it presents elements of specificity with less saturation of the g factor by the Process Indices and, above all, by the PSI; (b) with reference to performance levels, considering the classic four-factor structure of the WISC-IV, SLD present a typical profile with VCI and PRI (constitutive of the General Ability Index, GAI) clearly superior to WMI and PSI (constitutive of the Cognitive Proficiency Index, CPI); (c) this discrepancy can be used as support for the diagnosis; (d) a description of intelligence in terms of strengths and weaknesses and a reference to the GAI as a general index may be particularly appropriate in the case of SLDs; (e) within the population with SLDs, subgroups can be found with discrepancies between verbal and non-verbal performances, with different frequency and with peculiarities with respect to gender; (f) a bifactorial model confirms, in the case of SLDs, a predominance of specific factors with respect to the *g factor*.

## 5. Conclusions

The Italian Law n.170 of 8 October 2010 [30] recognizes dyslexia, dysorthography, dysgraphia, and dyscalculia as SLDs, assigning to the national education system and universities the task of identifying the most appropriate forms of teaching and assessment so that students with SLDs can achieve success.

The Guidelines present some directions, drawn up on the basis of the most recent scientific knowledge, to carry out individualized and personalized educational interventions, as well as to use compensatory tools and to apply dispensatory measures. They indicate the essential level of performance required of educational institutions and universities to guarantee the right to study of students with SLDs.

While affecting different abilities, the disorders described above can coexist in the same person, which is called "comorbidity". Success in learning is the immediate intervention to counteract the tendency of students with SLD to have poor self-esteem [31]. The cognitive specificity of students with SLDs also determines the consequences of the disorder at the school level, and important risk factors regarding repeated negative and frustrating experiences during the entire training process.

Students with SLDs show great psychological suffering related to the experiences of their deficiencies that can affect self-esteem and motivation and, on a relational level, can provoke problematic social functioning and/or a feeling of inferiority in interactions [32].

Strengthening, therefore, the pathways of diagnosis and assessment also means ensuring a greater understanding of the resources to be enhanced to support not only cognitive growth, but also emotional student relationships through strategic educational pathways tailored to specific needs [33].

**Author Contributions:** Conceptualization, D.D. and L.C.; methodology, D.D.; formal analysis, L.C.; resources, G.S.; data curation, L.C.; writing—original draft preparation, D.D.; writing—review and editing, D.D., L.C., G.S.; visualization, G.S. All authors have read and agreed to the published version of the manuscript.

**Funding:** This research received no external funding.

**Institutional Review Board Statement:** The study was conducted in accordance with the Declaration of Helsinki. The study was free of risks or charges, sponsors, conflicts of interest, and incentives for respondents. The study was conducted in accordance with the legislation of the Italian Code regarding the protection of personal data (Legislative Decree 196/2003); the participants were informed about the general purpose of the research, the anonymity of the answers, and the voluntary nature of participation, and they signed an informed consent form. There were no incentives given. The study

was conducted in accordance with D.lgs 101/2018 adeguated to GDPR UE 2016/679 (https://gdpr-info.eu/). Suitable recitals: (32) Conditions for Consent; (33) Consent to Certain Areas of Scientific Research; (38) Special Protection of Children's Personal Data; (40) Lawfulness of Data Processing (42); Burden of Proof and Requirements for Consent; (43) Freely Given Consent; (50) Further Processing of Personal Data; (51) Protecting Sensitive Personal Data; (54) Processing of Sensitive Data in Public Health Sector; (71) Profiling; (111) Exceptions for Certain Cases of International Transfers; (155) Processing in the Employment Context; (161) Consenting to the Participation in Clinical Trials; (171) Repeal of Directive 95/46/EC and Transitional Provisions. This study was approved by the independent commission of the "Centro di Counseling Psicologico" of the University of Salerno (Italy) (number 01/2019).

**Informed Consent Statement:** Informed consent was obtained from all subjects involved in the study.

**Data Availability Statement:** Written informed consent was obtained from the subject(s) in order to publish this paper.

**Conflicts of Interest:** The authors declare no conflict of interest.

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
