# Peer review of "Exploratory and Descriptive Comparison Using the WAIS-IV and LSC-SUA of the Cognitive Profile of Italian University Students with Reading Comprehension Disorder (RCD) vs. Decoding Difficulties—Dyslexia"

_2673-995X, doi:10.3390/youth3010023_

Round 1

Reviewer 1 Report

Introduction:

The introduction is clear and well writing. The objectives are clearly exposed.

Method section:

Lines 83 to 107: it is not clear if the assessment described concerns the method of this research or the classic assessment for student with special needs when he arrived at the University

Also, it is not clear when the Wechsler Adult Intelligence Scale was administered to participants

Results section:

Concerning this section, authors should be more specific about the analyses done and present the results of their inferential statistical analyses (regression for example). Currently, authors present only descriptive results and it is not sufficient. For example, is the difference between W_PRI in DD and RCD significant? Regarding the graphs (1 to 6), they are not necessary and are completely redundant with Table 2.

Discussion:

Important point, we don’t know if the discussion is supported by the results (i.e. comments on the results section).

Also:

Line 208-215: this paragraph must be rewritten.

More specifically:

Line 212: what is W_VIC? This score does not appear on the material description. Is it the W_VCI, if yes, then DD and RCD group had higher score on this one. Also, as W_GAI consists of the sub-tests W_VCI and W_PRI and W_ICC consists of W_WMI and W_PSI, it will be clearer to say if there is difference regarding these scores GAI and ICC between DD and RCD.

Line 214:  you wrote: “There is clearly a larger difference in the 214 scales W_GAI, W_ICC, W_VCI, W_WMI, and W_PSI”, but the W_PSI is not mentioned in the previous sentence.

Also, in the discussion, the authors could be more specific on the interest of their research.

Minors:

IN or INF used for inferential

Author Response

Introduction:

The introduction is clear and well writing. The objectives are clearly exposed.

Authors: Thank you for the positive feedback and for suggesting changes that improve our work.

Method section:

Lines 83 to 107: it is not clear if the assessment described concerns the method of this research or the classic assessment for student with special needs when he arrived at the University

Authors: The tests used for the study are the same that are used during the assessment in the course at the "Laboratory of the Study Method" of the University. We have clarified this point in the "Materials" section when we present the tests specifically.

Also, it is not clear when the Wechsler Adult Intelligence Scale was administered to participants

Authors: We have specified in the paragraph "Procedure" the moment of administration of WAIS-IV: “In the second and third meeting, the Reading, Writing, and Calculation tests of the LSC-SUA [12] and the WAIS-IV [10] are administered, which, depending on the student's difficulties and timeframe, may require another meeting”.

Results section:

Concerning this section, authors should be more specific about the analyses done and present the results of their inferential statistical analyses (regression for example). Currently, authors present only descriptive results and it is not sufficient. For example, is the difference between W_PRI in DD and RCD significant? Regarding the graphs (1 to 6), they are not necessary and are completely redundant with Table 2.

Authors: We removed the redundant graphs and rewrote the results. We also added a comparison with the normative group.

Discussion:

Important point, we don’t know if the discussion is supported by the results (i.e. comments on the results section).

Authors: We improved the "Discussions" section based on the results obtained.

Also:

Line 208-215: this paragraph must be rewritten.

More specifically:

Line 212: what is W_VIC? This score does not appear on the material description. Is it the W_VCI, if yes, then DD and RCD group had higher score on this one. Also, as W_GAI consists of the sub-tests W_VCI and W_PRI and W_ICC consists of W_WMI and W_PSI, it will be clearer to say if there is difference regarding these scores GAI and ICC between DD and RCD.

Line 214:  you wrote: “There is clearly a larger difference in the 214 scales W_GAI, W_ICC, W_VCI, W_WMI, and W_PSI”, but the W_PSI is not mentioned in the previous sentence.

Also, in the discussion, the authors could be more specific on the interest of their research.

Authors: We have completely remodulated the sentences indicated as specified. We also made the implications of our study clearer.

Reviewer 2 Report

Hello, this is an interesting subject but before publication is possible, fundamental changes are to be made. The manuscript, method and statistics are insufficiently described. Not sufficient literature was used. Now it is a very weak paper.

This are my suggestions:

I think adding a control group without dyslexia or RCD would heighten the meaningfulness of results.

Line 26) Is 19.616 the number of university students with dyslexia in Italy? Please specify. What might be the reason the number has grown (better support at school, more knowledge and diagnostic…)

Line 31) Why is it important which subject the students with dyslexia chose? Why do University’s in Italy know whether their students have dyslexia?

The theoretic outlet makes it not clear, why this study is important and conducted. None of the literature that is looking at specific and non-specific cognitive deficits in persons with SLD is cited. There are several studies that have shown links in the specific cognitive profile with the WISC-IV and a SLD. Why would you even suspect a relationship between specific cognitive profiles in the WAIS-IV and SLD? Please add those literature. Please also ass literature showing cognitive differences in persons with a reading disorder only and a dyslexia. Why is it RCD a specific disorder and not longer included in the dyslexia diagnosis?

Line 114 – the scale Index of cognitive competence not one of the primary four indices? The authors refer to the G-IQ aus T-IQ, please correct

Please give more information about the LSC-SUA and the subtests also include studies with the LSC-SUA.

Material and Methods

Procedure

In which time span was the data collected?

Line 127 Participants

Please bring in exact numbers – how many students were participating? On which criteria were the participating subjects chosen? Which test results in the LSC-SUA-Reading tests lead to inclusion of the RCD-Group and which in the dyslexia group? What were exclusion criteria? Please list descriptive data of the test results the participants gained in the WAIS-IV and the LSC-SUA-Reading tests. Did you include a discrepancy criterium to the G-IQ?

134 - Data analysis – when the groups were chosen according to the LSC-SUA-Reading tests – how can the result of the LSC-SUA-Reading test be the dependent variable without distorting the results?

Did the authors check whether there is no statistical correlation between the covarates and the independent variables?

Did the authors check whether the data meets the statistical prerequisites for computing a MANCOVA?

Why did the authors not include a control group?

Results

The authors only report descriptive statistics and presents a vast amounts of graphs that do not add information but hinder reading flow. Please report results of the MANCOVA and put the graphs in the attachment.

Table 1 – what does the W_ before the indices mean? Please specifiy

Table 2 – the given data is not informative as the used scale of the LSC-SUA Reading test are unknown. Please standardize the values (e.g. z-tranformation) or give further explanation. Please explan what Track A (TOT) Track A (INF) Track A (S) Track B (TOT) Track B (INF) Track B (S) means.

It would be here most informative to know about effect sizes of the difference of the scores to be able to judge wether the two performances differ and how strong.

Discusson:

As the results are presented inadequate, it is hard to follow the conclusions of the authors.

As the theoretic parts give no information about theories or previous studies with the WISC-WAIS and subjects with learning disorders, it is hard to follow the authors conclusions.

L. 208: The author refer to a new structure of the WAIS-IV – which new structure? The WAIS-IV is to my knowledge available in Italy since 2013 ( Orsini & Pezzuti, 20132015)

Line 225 ff: Regarding the second objective of our study, which was to examine, also in the Italian 225 context, which diagnostic materials can help to ascertain RCD in young adults, the tracks 226 of the LSC-SUA have been identified as a valuable tool for diagnostic evaluation, includ- 227 ing discriminating the performance of students with RCD and DD. In fact, from the anal- 228 ysis of the average scores obtained on the comprehension of the LSC-SUA tracks, the DD 229 Group had higher average scores than the RCD Group on both Track A and B in all the 230 characterizing tests.

I don’t see why the authors could possibly draw such a conclusion – by descriptive analysis of the data only? How were the groups selected?

The authors need to fully and carefully revise the paper. 

Author Response

Hello, this is an interesting subject but before publication is possible, fundamental changes are to be made. The manuscript, method and statistics are insufficiently described. Not sufficient literature was used. Now it is a very weak paper.

Authors: Thank you for your suggestions to improve our work.

This are my suggestions:

I think adding a control group without dyslexia or RCD would heighten the meaningfulness of results.

Authors: Ours is an exploratory study to improve university services and better meet the needs of students. Being a preliminary study, we compared our results with the normative group.

Line 26) Is 19.616 the number of university students with dyslexia in Italy? Please specify. What might be the reason the number has grown (better support at school, more knowledge and diagnostic…)

Line 31) Why is it important which subject the students with dyslexia chose? Why do University’s in Italy know whether their students have dyslexia?

Authors: We specified: In Italy, the number of university students with Dyslexia and other Specific Learning Disabilities (SLD) is growing: in 2020–2021, there were 19.616, an increase of 22% compared to the previous academic year. According to research carried out by the national agency for the evaluation of the university and research system in Italy [1], 54% of university students with SLD chose socio-humanistic disciplines, 28.7% chose scientific areas, and 9.4% chose health. Approximately 13.127 students with SLD were enrolled in three-year degree courses, 1.507 in Master's degree courses, and 1.416 in single-cycle courses. A total of 90 out of 98 universities participated in the survey: 76.7% offered specific guidance services (before, during, and after the study course), and 68.7% provided support for teaching, including specialised or peer tutoring and the provision of books in accessible format. Clearly, understanding and recognising SLD at university is essential to enable students to access tutoring services and to be able to complete their studies to the best of their ability [1].

The theoretic outlet makes it not clear, why this study is important and conducted. None of the literature that is looking at specific and non-specific cognitive deficits in persons with SLD is cited. There are several studies that have shown links in the specific cognitive profile with the WISC-IV and a SLD. Why would you even suspect a relationship between specific cognitive profiles in the WAIS-IV and SLD? Please add those literature. Please also ass literature showing cognitive differences in persons with a reading disorder only and a dyslexia. Why is it RCD a specific disorder and not longer included in the dyslexia diagnosis?

Authors: We improved both the "Results" and "Discussion" sections by highlighting the comparison between the groups and the implications of our study.

Line 114 – the scale Index of cognitive competence not one of the primary four indices? The authors refer to the G-IQ aus T-IQ, please correct

Authors: Done

Please give more information about the LSC-SUA and the subtests also include studies with the LSC-SUA.

Authors: Done

Material and Methods

Procedure

In which time span was the data collected?

Authors: academic years 2020-2021. Inserted the information.

Line 127 Participants

Please bring in exact numbers – how many students were participating? On which criteria were the participating subjects chosen? Which test results in the LSC-SUA-Reading tests lead to inclusion of the RCD-Group and which in the dyslexia group? What were exclusion criteria? Please list descriptive data of the test results the participants gained in the WAIS-IV and the LSC-SUA-Reading tests. Did you include a discrepancy criterium to the G-IQ?

Authors: We have remodulated the section of “Results”.

134 - Data analysis – when the groups were chosen according to the LSC-SUA-Reading tests – how can the result of the LSC-SUA-Reading test be the dependent variable without distorting the results?

Did the authors check whether there is no statistical correlation between the covarates and the independent variables? Did the authors check whether the data meets the statistical prerequisites for computing a MANCOVA?

Authors: as also suggested by the other reviewer we have eliminated the redundant part of multivariate analysis. We focused on describing the results and comparing normative data.

Why did the authors not include a control group?

Authors: We have inserted the comparison of the group of our study with that normative one.

Results

The authors only report descriptive statistics and presents a vast amounts of graphs that do not add information but hinder reading flow. Please report results of the MANCOVA and put the graphs in the attachment.

Authors: also suggested by the other reviewer we have eliminated the redundant part of multivariate analysis.

Table 1 – what does the W_ before the indices mean? Please specifiy

Authors: “W” is eliminated. We add footnote for the table.

Table 2 – the given data is not informative as the used scale of the LSC-SUA Reading test are unknown. Please standardize the values (e.g. z-tranformation) or give further explanation. Please explan what Track A (TOT) Track A (INF) Track A (S) Track B (TOT) Track B (INF) Track B (S) means.

It would be here most informative to know about effect sizes of the difference of the scores to be able to judge wether the two performances differ and how strong.

Authors: We added specific for the table and insert specification of the analysis.

Discusson:

As the results are presented inadequate, it is hard to follow the conclusions of the authors.

As the theoretic parts give no information about theories or previous studies with the WISC-WAIS and subjects with learning disorders, it is hard to follow the authors conclusions.

Authors: We improved “Discussion” section.

  1. 208: The author refer to a new structure of the WAIS-IV – which new structure? The WAIS-IV is to my knowledge available in Italy since 2013 ( Orsini & Pezzuti, 2013, 2015)

Authors: “new” was eliminated.

Line 225 ff: Regarding the second objective of our study, which was to examine, also in the Italian 225 context, which diagnostic materials can help to ascertain RCD in young adults, the tracks 226 of the LSC-SUA have been identified as a valuable tool for diagnostic evaluation, includ- 227 ing discriminating the performance of students with RCD and DD. In fact, from the anal- 228 ysis of the average scores obtained on the comprehension of the LSC-SUA tracks, the DD 229 Group had higher average scores than the RCD Group on both Track A and B in all the 230 characterizing tests.

I don’t see why the authors could possibly draw such a conclusion – by descriptive analysis of the data only? How were the groups selected?

Authors:  We have completely remodelled this part. Both in the results and in the discussion.

The authors need to fully and carefully revise the paper.

Authors: 

With regard to the linguistic revision, we rewrote to the MDPI English editing Editor, who had provided us with the certification of the translation and made some very small revisions

Round 2

Reviewer 1 Report

Method section:

Lines 83 to 107: it is still not clear if the assessment described concerns the method of this research or the classic assessment for student with special needs when he arrived at the University

More precisely, it seems that the Wechsler Adult Intelligence Scale and the LSC-SUA were administered twice: during the assessment from the “Disabilities/SLD Help Desk” and during the assessment at the “Laboratory of the Study Method”

Results section:

My previous comments seem not to have been taken into account: authors should be more specific about the analyses done and present the results of their inferential statistical analyses. Currently, authors still present only descriptive results and it is not sufficient to conclude on difference on scores. For example, is the difference between W_PRI in DD and RCD significant?

Discussion:

Important point, we still don’t know if the discussion is supported by the results (i.e. comments on the results section).

Also, in the discussion, the authors could be more specific on the interest of their research.

Author Response

Method section:

Lines 83 to 107: it is still not clear if the assessment described concerns the method of this research or the classic assessment for student with special needs when he arrived at the University

More precisely, it seems that the Wechsler Adult Intelligence Scale and the LSC-SUA were administered twice: during the assessment from the “Disabilities/SLD Help Desk” and during the assessment at the “Laboratory of the Study Method”

Authors: we have removed the wording “Laboratory of the Study Method”  and used only “Disabilities/SLD Help Desk” as it was misleading. The assessment methodology of the “Disabilities/SLD Help Desk” is that of the research protocol.

Results section:

My previous comments seem not to have been taken into account: authors should be more specific about the analyses done and present the results of their inferential statistical analyses. Currently, authors still present only descriptive results and it is not sufficient to conclude on difference on scores. For example, is the difference between W_PRI in DD and RCD significant?

Authors: we have entered levels of significance.

Discussion:

Important point, we still don’t know if the discussion is supported by the results (i.e. comments on the results section).

Also, in the discussion, the authors could be more specific on the interest of their research.

Authors: we have improved the 'Discussion' paragraph as suggested.

Reviewer 2 Report

Line 145 please follow APA guidelines to report statistics (https://apastyle.apa.org/instructional-aids/numbers-statistics-guide.pdf) (M; SD – decimal places)(12 M; mean age= 21.5; SD= 22) – how can the SD be 22is that years?

English:

Line 28 there were 19.616 – missing subject

Line 147 during di academic years 2020-2021.

n°10 students with a diagnosis of RCD (RCD Group) and n°10 with De- 148 coding Difficulties—Dyslexia (DD Group)

How did the participants got selected – by which cut off? Were there exclusion criteria? Were they controlled for other diagnosis (e.g. ADHD)

The authors still refer to the G-IQ as TIQ (L162) WAIS-IV Index: Total Intelligence Quotient (TIQ) despite of having said replacing it

Table 1 – the authors report SD to the third decimal – this is not according to APA

The authors only report descriptive statistics. No inferences can be drawn from presenting mere means and standard deviations. If the whole approach is only explorative and descriptive, the authors must mark this in the title and abstract.

Still, there is no information how the two groups were chosen – which cut off was used? So, this paper is not ready for publication.

A number of issues I mentioned was not addressed (incuding literature about different cognitive profiles of individuals with reading/decoding disorder)

Author Response

Line 145 please follow APA guidelines to report statistics (https://apastyle.apa.org/instructional-aids/numbers-statistics-guide.pdf) (M; SD – decimal places)(12 M; mean age= 21.5; SD= 22) – how can the SD be 22 – is that years?

Authors: use APA stile for define it

English:

Line 28 there were 19.616 – missing subject

Line 147 during di academic years 2020-2021.

Authors: Done

n°10 students with a diagnosis of RCD (RCD Group) and n°10 with De- 148 coding Difficulties—Dyslexia (DD Group)

How did the participants got selected – by which cut off? Were there exclusion criteria? Were they controlled for other diagnosis (e.g. ADHD)

Authors: We included the Vinegrad Plus cut-off level and the exclusion of other diagnoses.

The authors still refer to the G-IQ as TIQ (L162) WAIS-IV Index: Total Intelligence Quotient (TIQ) despite of having said replacing it

Authors: We focused more on the profiles that emerged from the WAIS-IV

Table 1 – the authors report SD to the third decimal – this is not according to APA

Authors: Done

The authors only report descriptive statistics. No inferences can be drawn from presenting mere means and standard deviations. If the whole approach is only explorative and descriptive, the authors must mark this in the title and abstract.

Authors: Done

Still, there is no information how the two groups were chosen – which cut off was used? So, this paper is not ready for publication.

Authors: we have entered the specifications of the Vinegrag Plus cut-off and inclusion criteria

A number of issues I mentioned was not addressed (incuding literature about different cognitive profiles of individuals with reading/decoding disorder)

Authors: We have added changes based on suggestions in the Discussions section

Round 3

Reviewer 1 Report

Introduction:

Method section:

Authors answered to my comments.

Results section:

Authors added some p values, but we don’t know what statistical test was used. More problematic, we don’t know to which comparisons the tests refer to! Finally, if several tests were done, which correction was applied?

One more time, authors should be more specific about the analyses done and present the results of their inferential statistical analyses correctly

Discussion:

Important point, we still don’t know if the discussion is supported by the results (i.e. comments on the results section).

Author Response

Method section:

Authors answered to my comments.

Authors: thank you for your feedback and suggestions!!

 Ì´

Results section:

Authors added some p values, but we don’t know what statistical test was used. More problematic, we don’t know to which comparisons the tests refer to! Finally, if several tests were done, which correction was applied?

One more time, authors should be more specific about the analyses done and present the results of their inferential statistical analyses correctly

Authors: We have detailed the section "Data analysis" and carried out significance tests again. We have also included correlation analyses. We have also modified Tables 1 and 2.

Discussion:

Important point, we still don’t know if the discussion is supported by the results (i.e. comments on the results section).

Authors: We have included additional studies in the literature that confirm our results. They have been detailed in the "Discussions" section.

Reviewer 2 Report

In the third round of revision, there are still important points that needs to be worked on bevor this paper can be published. Here are my comments:

In the abstract, the authors use the therm TIQ and ICC to refer to WAIS – scales. The right therm as referred to by Wechsler to address the “total IQ” is Full Scale IQ – please refer to it as FSIQ – what is meant by ICC?

In the theoretical questions, you line up the question: To investigate the relationship between cognitive 79 abilities assessed with the Wechsler Adult Intelligence Scale—Fourth Edition (WAIS-IV) 80 [10] in adult students with RCD in comparison with students with Decoding Disorder 81 (DD);

But you don’t present any of the quite large body of literature that there is considering differences in the cognitive profile between the different subtypes of learning disorders. Please include

Materials – please refer to the common therms when talking about the WAIS, e.g. Full Scale IQ

Participants

Now you included information about how the both subgroups are selected: both subgroups, students were selected on the basis of their score on the Vine- 150 grad Plus (cut-off = 11), which allows for 73% positivity of diagnosis, with a 5% probability 151 of false positives [12].

Still, from this explanation, it is not clear how you did differ between the DD and the RCD group – as I read it now, I can understand how you sample out students that have a higher risk of a dyslexia in this questionnaire

Results

You report significant differences between the groups, but from the description I can’t understand which statistics did you use? Please report the statistics according to APA, also with effect sizes!

Also, which statistics program did you use? What was you alpha-level set up to? Which models did you set up? Please add a section: statistical analysis

As long as the statistics are not sufficiently explained, I can’t make a useful revision of the discussion part

Author Response

In the third round of revision, there are still important points that needs to be worked on bevor this paper can be published. Here are my comments:

Authors: thank you for your feedback and suggestions!!

In the abstract, the authors use the therm TIQ and ICC to refer to WAIS – scales. The right therm as referred to by Wechsler to address the “total IQ” is Full Scale IQ – please refer to it as FSIQ – what is meant by ICC?

Authors: We have changed TIQ to FSIQ and ICC to CPI not only in the abstract but also in the main paper and table. FSIQ is the index of cognitive competence as described in the Wechsler Intelligent Scale for Children (WISC-IV; Wechsler, 2003) and described in Materials and Methods. Furthermore, we have detailed that “In addition, in accordance with the study by Cornoldi et al. [13], we used two additional composite indices: General Ability Index (GAI, consisting of the sub-tests VCI and PRI) and Cognitive Proficiency Index (CPI, consisting of WMI and PSI)”.

In the theoretical questions, you line up the question: To investigate the relationship between cognitive 79 abilities assessed with the Wechsler Adult Intelligence Scale—Fourth Edition (WAIS-IV) 80 [10] in adult students with RCD in comparison with students with Decoding Disorder 81 (DD);

But you don’t present any of the quite large body of literature that there is considering differences in the cognitive profile between the different subtypes of learning disorders. Please include

Authors: We have included additional studies in the literature that confirm our results. They have been detailed in the "Discussions" section. Furthermore, we have modified aim  (a) of the study.

Materials – please refer to the common therms when talking about the WAIS, e.g. Full Scale IQ

Authors: We have changed TIQ to FSIQ

Participants

Now you included information about how the both subgroups are selected: both subgroups, students were selected on the basis of their score on the Vine- 150 grad Plus (cut-off = 11), which allows for 73% positivity of diagnosis, with a 5% probability 151 of false positives [12].

Still, from this explanation, it is not clear how you did differ between the DD and the RCD group – as I read it now, I can understand how you sample out students that have a higher risk of a dyslexia in this questionnaire

Authors: We have included the cut-off for Viegrad Plus which allows for an initial screening. We have also included specifications for the general inclusion criteria of the study and specific for the two groups of interest (DD and RCD).

Results

You report significant differences between the groups, but from the description I can’t understand which statistics did you use? Please report the statistics according to APA, also with effect sizes!

Also, which statistics program did you use? What was you alpha-level set up to? Which models did you set up? Please add a section: statistical analysis

As long as the statistics are not sufficiently explained, I can’t make a useful revision of the discussion part

Authors: We have detailed the section "Data analysis" and carried out significance tests again. We have also included correlation analyses. We have also modified Tables 1 and 2.